# Evaluation of Apoptotic Caspase-3 Immunopositivity in Human Model of Asphyxial Death

**DOI:** 10.3390/ijms26073317

**Published:** 2025-04-02

**Authors:** Fabio Del Duca, Michele Treglia, Raffaele La Russa, Stefania De Simone, Luigi Cipolloni, Aniello Maiese, Paola Frati

**Affiliations:** 1Department of Anatomical, Histological, Forensic and Orthopedic Sciences, Sapienza University of Rome, Viale Regina Elena 336, 00161 Rome, Italy; fabio.delduca@uniroma1.it (F.D.D.); paola.frati@uniroma1.it (P.F.); 2Department of Biomedicine and Prevention, University of Rome “Tor Vergata”, 00161 Rome, Italy; michelemario@hotmail.it; 3Department of Clinical Medicine, Public Health, Life Sciences, and Environmental Sciences, University of L’Aquila, 67100 L’Aquila, Italy; raffaele.larussa@univaq.it; 4Department of Clinical and Experimental Medicine, University of Foggia, 71100 Foggia, Italy; stefania.desimone@unifg.it (S.D.S.); luigi.cipolloni@unifg.it (L.C.)

**Keywords:** hanging, asphyxia, vitality, autopsy, postmortem, ligature mark, forensic

## Abstract

The pathological mechanisms underlying the ligature mark in hanging involve the skin layers and an ischemic mechanism. The apoptotic process develops whenever ischemic mechanisms affect the dermal and epidermal layers. Effector caspase-3 appears to play a crucial role in both acute and chronic pressure-induced skin ischemia. The aim of this study is to identify the role of caspase-3 as a marker of supravitality in the diagnosis of premortem hanging. Skin samples from ligature marks in hanging cases were collected to investigate this apoptotic process. The caspase-3 levels in compressed skin were significantly higher compared to those found in healthy skin (*p* < 0.005). The apoptotic process in ischemic epidermal cells begins with stable mechanical stress, as seen in the hanging model. Caspase-3 expression seems to vary from minutes after the initial stress input. Caspase-3 activation is an ATP-dependent process and can only occur if the victim was alive before the pressure was applied. Caspase-3 is a reliable marker of supravitality in ligature marks in premortem hanging cases.

## 1. Introduction

Caspase-3, a protein crucial to apoptotic processes, is encoded by the CASP3 gene. Functioning as a cysteine-aspartic protease, it employs cysteine as a catalytic nucleophile within its active site. This enables caspase-3 to cleave target proteins precisely at specific aspartic acid residues, initiating a caspase cascade [1].

Laboratory studies on engineered skin samples subjected to sustained high pressure via hydrostatic force have demonstrated an increase in programmed cell death (apoptosis). This phenomenon was confirmed by the presence of the anti-caspase-3 antibody in the compressed tissues, detected through immunohistochemical analysis [2].

Apoptosis is an ATP-dependent programmed cell death triggered by various stressors, including ischemia, trauma, and exposure to reactive oxygen species [3]. In forensic investigations, the postmortem detection of apoptosis in cells suggests a supravital phenomenon [4]. Despite the irreversible cessation of vital functions and the onset of ischemia, residual ATP may still be present, enabling the apoptotic cascade to continue. This indicates that apoptosis can persist in ischemic cells until energy reserves are fully depleted. The identification of apoptotic activity in compressed tissues may serve as a potential marker of vitality.

This study aims to investigate the response of furrowed skin in cases of hanging and to determine whether caspase-3 can serve as a marker of vitality.

In deaths by hanging, skin tissue undergoes sustained compression during the agonal phase. To establish the circumstances surrounding a hanging death, forensic analysis must identify markers of tissue response to the compressive trauma induced by the ligature [5].

Hanging is generally presumed to be a suicidal act, though accidental cases can occur, particularly in work-related settings where safety devices are involved. Homicidal hanging is rare but possible. Although homicide rates are relatively high, it is crucial to acknowledge that homicidal hangings do occur [6].

The reported prevalence of homicidal hanging varies significantly across studies. A review of 1500 cases identified only one homicide [6], while another study reported a 1.6% incidence of homicides by hanging or staged hangings [7,8,9,10,11,12,13,14,15].

Two primary categories of homicidal hanging have been described: true homicidal hanging and simulated hanging. True homicidal hanging occurs when asphyxia due to hanging directly causes death. Simulated hanging, by contrast, refers to cases where the victim is killed by other means before being placed in a noose to mimic suicide. This scenario is often observed in drug-related deaths, where the body is suspended by the neck postmortem to conceal an unintentional homicide [6].

## 2. Results

Microscopic examination of skin slides from 21 hanging cases revealed, corresponding to the skin compression groove, flattening and compression of the epidermal layer with the formation, in some areas, of intra-epidermal vesicles filled with fluid. In some sections, it was also possible to observe some leukocytic cells in the dermal layer and characteristics interpretable as Zenker’s necrosis at the level of the subcutaneous muscle layer.

In cases of postmortem cadaver suspension, metachromasia of the connective tissue and stasis at the vascular level were observed, with substantial normality at the muscular level.

In samples of healthy skin, normal alternation of the skin layers was observed, which showed no significant alterations, as well as substantial normality at the muscular level.

Caspase-3 protein, present within epidermal cells, is more strongly expressed in the basal layer (Figure 1A–D). At this level, epidermal cells within the ligature furrow appear flattened but remain histologically identifiable. Cleaved caspase-3 is distinctly visible with a cytoplasmic distribution, exhibiting a granular pattern, and is also present within the nucleus (Figure 2A,B). The distribution of caspase-3 is evident in the dermis as well, with focal positivity observed in dermal vessels and fibroblasts, consistent with findings from similar studies.

A semi-quantitative analysis conducted in this study revealed an overexpression of caspase-3 in epidermal cells associated with areas of epidermal flattening, homogenization of epidermal cells, and detachment of the stratum corneum.

The mean difference in intensity value is 2.48 ± 0.51 SD (IQ1 2 IQ3 3; variance 0.26). The surrounding epidermal cells of the healthy skin showed an almost uniform underexpression in all samples (see Table 1).

Uninjured skin surrounding the ligature mark showed a significantly lower semi-quantitative evaluation (mean intensity value 0.23 ± 0.44 SD). The caspase-3 values of the compressed skin were significantly increased compared to the values found on healthy skin (*p* < 0.005).

## 3. Discussion

This study was conducted within a research group that has long been interested in studying the vitality of lesions, specifically those attributable to the ligature mark [16], which is useful for a diagnostic assessment [17,18,19].

The physio-pathological mechanisms underlying the ligature mark in the case of hanging are attributable to dehydration of the skin of the furrow, inflammation and apoptosis. Ishida et al. (2018) highlighted the role of aquaporins in the diagnosis of vitality of the skin of the antemortem hanging furrow [20].

The AQP family consists of channel proteins linked to the cell membrane, composed of 13 homologous proteins. AQPs transport water, glycerol or other small solutes. The aquaporins involved in the study by Ishida et al. are the AQP3s, which are essential for water reabsorption and, therefore, for the correct homeostasis of epidermal cells.

AQP3 was overexpressed on the keratinocytes of antemortem hanging ligature marks. Given that the skin of the ligature furrow is subjected to constant compression, and shows strong aspects of dehydration upon macroscopic examination, it is possible to hypothesize that AQP3 is expressed in response to dehydration. In summary, the transdermal water loss—a measure of the permeability barrier in the cutaneous epidermis—could elicit a cellular response mediated by the overexpression of AQP3, in order to replace the water loss.

Regarding dehydration, there is also a hydro-electrolytic commitment of the cytosol, as revealed by Pérez et al., which showed an increase in intracellular Fe and Zn which was correlated with elements of inflammatory interest such as cathepsin D and P-selectin [21].

Inflammation is, however, certainly elicited by the inflammatory stimulus, as most of the immunohistochemical studies on the skin of the furrow have shown.

The key study by Turillazzi et al. (2010) highlighted the overexpression of molecules related to innate inflammatory cell hyperactivation, particularly those related to neutrophils and cell recruitment [22]. From the immunohistochemical study, a statistically significant positivity for IL-15, CD15 and tryptase was revealed: IL-15, localized around the dermal vessels and the subcutaneous connective tissue, while CD15 and tryptase were expressed in the dermal connective tissue.

Further studies have analyzed the structural and functional alterations of epithelial cells, and the immunological markers associated with flogosis in response to mechanical compression of the neck. According to Dudnyk, Focardi et al. (2022), Zhang et al. (2023), and Caputo et al. (2023), numerous markers were detected that gave statistically significant evidence regarding the diagnosis of furrow vitality, such as IL-1β, ubiquitin, fibronectin, P-Selectin, FVIII, MRP-8 [23,24,25,26].

These are markers that are involved in the cell adhesion and migration phase in the case of an acute inflammatory response, as in the case of fibronectin and P-selectin, which are useful for cell recruitment, elicited by inflammatory cytokines such as IL-1β. The molecular expression of genetic material was also investigated by Neri et al. (2019) [27,28].

With this study, they brought to light the overexpression of miRNA (miR103a-3p, miR214-3p, and miR92a-3p) correlated with inflammation at the level of the ligature furrow. The correlation was shown to be statistically significant and can be considered as potential targets for the diagnosis of ligature furrow vitality.

Maiese et al. (2020) studied c-FLIP, an inhibitor of apoptosis, and its pathophysiological underexpression in cases of suicidal hanging [29]. In the study group, a depletion of intracytoplasmic c-FLIP was reported, while no hypoexpression was found in the control group; the differences between the groups were statistically significant.

In a further analogous work, Maiese et al. (2023) focused on the expression of FOXO3 to evaluate whether its depletion could support the diagnosis of vitality in ligature marks [16].

The skin samples were analyzed using immunohistochemical techniques with specific antibodies for FOXO3 and a semi-quantitative immunocoloration analysis was performed.

The results showed a statistically significant reduction in the expression of FOXO3 in skin samples collected from ligature marks compared to uninjured skin and postmortem samples, highlighting an intracytoplasmic and nuclear localization, according to a previous study.

As noted in previous studies, the ligature mark exhibits characteristics of compression and ischemia, which may elicit an apoptotic response of both intrinsic and extrinsic nature.

In particular, caspase-3 is activated by the cleavage of pro-caspase-3 by the apoptosome, both in the extrinsic pathway (death receptor—caspase 8) and in the intrinsic pathway (mitochondrial caspase-9) [30]. The immunohistochemical study of caspase-3 has been conducted in other forensic pathology cases, both in murine models and in humans. In rats methamphetamine abuser, an increased expression of caspase-3 has been observed in the prefrontal cerebral cortex [31].

In other studies, caspase-3 has been evaluated for the estimation of the postmortem interval. The initiator caspases (8 and 9) and effector caspases (3, 6 and 7) are activated during apoptosis. The caspase-3 mRNA level significantly increased (*p* < 0.05) on injured rats’ skin [32].

In the present study, the presence of an overexpression of caspase-3 was demonstrated, with a statistically significant increase compared to healthy skin. The caspase-3 values of the compressed skin were shown to be statistically significantly increased compared to the values found on healthy skin (*p* < 0.005).

Comparing previous studies, it is evident that the underexpression of c-FLIP and FOXO3 and the overexpression of caspase-3 should be considered as mirror opposites.

The molecular pathophysiological interpretation is hypothesized to involve the induction of apoptosis through the downregulation of c-FLIP. The decrease in c-FLIP, compromising its anti-apoptotic function, may lead to the activation of caspase-3.

Numerous transcription factors trigger the transcription of c-FLIP, including NF-κB, AP-1 (cFos/c-Jun), P53, FoxO, CREB, NFATc2, EGR1, AR, and SP1 [33,34,35,36,37]. Others induce the repression of c-FLIP expression, such as c-Fos, c-Myc, FoxO3a, IRF5, and SP3 [38,39].

From the previous studies, it should be prudently assessed whether the nuclear translocation of FOXO3a is regulated by Akt, which, through phosphorylation, prevents its entry into the nucleus from the cytoplasm. Conversely, the non-phosphorylated form of FOXO3 enables its nuclear translocation, leading to the downregulation of FLIP levels and the production of pro-apoptotic proteins (such as BIM) [16,40].

Thus, the presence of the apoptotic process within the compressed skin of the ligature furrow is confirmed by the overexpression of caspase-3. This finding is consistent with the pro-apoptotic role of nuclear FOXO3 (which is hypoexpressed) and the loss of cytoplasmic c-FLIP (Figure 3).

It can, therefore, be concluded that skin ligation in hanging cases activates the intrinsic apoptotic pathway, mediated by the production of reactive oxygen species due to ischemia in the ligature furrow.

One of the main limitations of this study is the small sample size, which prevents proper stratification and validation of the immunohistochemical data. Selection biases include interpersonal variability, differences in sample sizes, and variations in hanging methods. Additionally, inter-operator variability may have influenced the assessment of positivity and negativity in the semi-quantitative analysis of the immunohistochemical preparations.

## 4. Materials and Methods

### 4.1. Study Group Selection and Samples Collection

The autopsy databases from the Legal Medicine and Forensic Institutes of “Sapienza” University of Rome and the University of Foggia were consulted retrospectively. After reviewing autopsy reports and information gathered from police investigations, 21 cases of suicidal hanging deaths were chosen for the study.

Between 2020 and 2024, the autopsy databases of the Legal Medicine and Forensic Institutes of Sapienza University of Rome and the University of Foggia recorded cases of autopsies for all causes of death, both violent and natural. From this dataset, cases were randomly selected from autopsies ordered by judicial authorities in cases of suspected criminal activity. Once anonymized, the database was categorized into violent and non-violent deaths. The violent death cases were analyzed, and a subset of individuals who died by hanging was extracted. The study group was then selected from this database through a random selection of 21 cases. The primary endpoint of the study was to assess the vitality of the ligature mark in all hanging cases, regardless of the knot position. The same procedure was applied to select the control cases.

Cases involving decomposed bodies, even with initial signs of putrefaction, or those lacking clear details about the circumstances of death were not included. The study group consisted of 8 women and 13 men, with an average age of 52.2 years. In 11 of the 21 cases, the individuals used wide, soft, and flexible materials like sheets for hanging, while in the other 10 cases, they used hard materials like ropes. A total of 17 cases out of 21 were classified as typical hangings, as reported by Chacko et al. 2025 [41]. In fact, the knot was positioned at the back of the neck in 7 cases, on the right side in 10 cases, and on the left side in 4 cases. The hanging was complete in 15 cases and incomplete in 6 cases.

For comparison, a control group of 21 individuals was selected, comprising 10 women and 11 men with an average age of 47.3 years. Samples of seemingly healthy skin were taken from these individuals at a later time. Cases with decomposed bodies or initial signs of putrefaction were excluded. The causes of death in the control group were drug overdoses in 8 cases, car accidents in 4 cases, and sudden cardiac deaths in 8 cases. In 3 of the 8 drug overdose cases, the bodies were suspended after death.

All bodies underwent autopsy within 36 h of death. During the autopsy, skin samples were collected from each case. In both antemortem and postmortem hanging cases, skin sections were taken from the neck where the mark was deepest. In contrast, skin samples from the control group were taken from the front of the neck.

This study represents an implementation of our previous studies, and the study group and skin samples are the same as described in our previous work on apoptosis molecule expression in the ligature mark [16,29].

The processing of the data reported in this paper is covered by the general authorization to process personal data for scientific research purposes granted by the Italian Data Protection Authority (1 March 2012 as published in Italy’s Official Journal no. 72 dated 26 March 2012) because the data do not entail any significant personalized impact on the data subjects. Our study did not involve the application of experimental protocols; therefore, it did not require approval by an institutional and/or licensing committee. The bodies included in this study were autopsied by order of the Italian Judicial Authority. In all cases, local prosecutors opened an investigation, ordering an autopsy to be performed to clarify the exact cause of death. Therefore, according to the Italian law, no ethical approval was needed. However, the present study was conducted with respect for the deceased involved and data were anonymized to guarantee the privacy of each subject.

### 4.2. Histological and Immunohistochemical Analysis

A routine microscopic histopathological study was performed using hematoxylin-eosin (H and E) staining. In addition, an IHC investigation of skin samples was performed, as previously published.

Skin fragments measuring 0.5 cm × 0.8 cm were collected. Samples, 8 cm^2^, from each case were fixed in 10% buffered formalin and then washed with phosphate-buffered saline (PBS), and subsequent dehydration was carried out using a graded alcohol series. The fixation time in formalin varied from a minimum of 7 h to a maximum of 72 h. After dehydration, samples were cleaned in xylene and embedded in paraffin. Sections were cut at 4 μm, mounted on slides, and covered with 3-amminopropyltriethoxysilane (Fluka, Buchs, Switzerland).

To evaluate the immunohistochemical positivity of the caspase-3 antibody, the mouse anti-caspase-3 antibody produced by Santa Cruz-Dallas, Texas U.S.A (SC-7272) was used. The antigen unmasking process was carried out using an EDTA solution in a microwave oven for 10 min. After washing with PBS, the blocking serum (bovine serum albumin) was incubated for an additional 15 min to prevent possible non-specific binding of the primary antibody.

The primary anti-caspase-3 antibody (SC-7272) was diluted at a ratio of 1:100 in the same blocking serum, in a sufficient amount to obtain a solution that allowed the incubation of about 100 µL of diluted solution per sample.

At the end of the incubation with the primary antibody, the slide was washed with PBS and incubated for 15 min with a secondary anti-goat biotinylated antibody from a different animal species from the type of antibody selected.

At the end of the second incubation, the biotin-avidin bond was obtained by incubating the sample at the site of the immunocomplex with streptavidin bound to a catalyst (peroxidase) for a further 15 min. After further washing, in order to detect the occurrence of antibody binding, di-amino-benzidine (DAB) was chosen as the chromogen for the time necessary to evaluate the reaction.

Subsequently, the sample was subjected to staining with Meyer’s hematoxylin for 30 s in order to proceed with the staining of the ultra-cellular basophilic components (nucleus). The further dehydration of the sample was achieved by reverse passage through the alcohol series and diaphanization in xylene with subsequent mounting with Eukit.

### 4.3. Quantitative Analysis

As described in our previous study, each immunohistochemical slide underwent a quantitative analysis. For quantitative analysis, in each immunohistochemical section, 20 observations were made in different fields/slides at 100-fold magnification. The samples were also examined under a confocal microscope and a three-dimensional reconstruction was performed (True Confocal Scanner, Leica TCS SPE, Cambridge, UK).

The staining intensity was evaluated using a semi-quantitative scoring scale. A semi-quantitative blind evaluation of the IHC findings by two different investigators (AM and FDD) was performed. All measurements were carried out at the same magnification of image (×10) and the gradation of the immunohistochemical reaction was assessed using a scale from 0 to 3, as shown in Table 2. The grade was based on the maximum expression of caspasi-3 noted. The evaluations were carried out separately for each sample using a double-blind method. In cases of divergent scoring, a third observer (ET) decided the final score.

### 4.4. Statistical Analysis

Semi-quantitative evaluation of the IHC findings and gradation of the immunohistochemical reaction were described using an ordinal scale. The median values were then reported. Analysis of variance for the non-parametric data was performed using a Kruskal–Wallis test. When differences were found to be significant, analysis between the unmatched groups was elucidated using Dunn’s multiple comparison post hoc test. The significance level was set to 5% (SPSS ver. 16.01 for Windows—SPSS Inc., Chicago, IL, USA).

## 5. Conclusions

The investigation of the immunohistochemical distribution of molecules related to vitality assessment is developing as a gold standard in the diagnosis of hanging. In medico-legal autopsy cases of suspected hanging, death is a clue to the presence of a peculiar blunt injury caused by a rope, string or a cord around the victim’s neck.

Despite variabilities, unique characteristics are displayed on cervical injured skin, including tiny epidermal fragments and a yellowish firmness, resulting from dehydration. It represents a *limine vitae* occurrence, sometimes observed in injuries occurring after death.

Therefore, it is crucial to identify molecular indicators that can help distinguish wounds sustained before death in such instances. Detecting *supervitality markers* is essential, as they are activated at the initial stage of trauma and exhibit immunoreactivity within minutes. For this reason, numerous methods and biomarkers are currently being investigated.

Moreover, demonstrating the vitality of a wound during the post-traumatic interval is challenging due to supravital reactions and morphological changes that occur over time.

As previously published, the role of mechanical stress-induced apoptosis has been investigated by revealing the role of caspase-3 in injured skin of the neck in hanging cases. Caspase-3 is involved in the last phase of the apoptosis process, resulting in the breakdown of DNA, the disintegration of cytoskeletal and nuclear proteins, the formation of protein cross-links, the emergence of apoptotic bodies, the presentation of ligands for phagocyte receptors, and, eventually, their engulfment by phagocytic cells.

In cervical injured skin in cases of hanging, the immunohistochemical distribution of caspase-3 seems to be a reliable marker in medicolegal autopsy cases. Caspase-3 should be considered for the postmortem investigation of stress responses to premortem hanging.

## Figures and Tables

**Figure 1 ijms-26-03317-f001:**
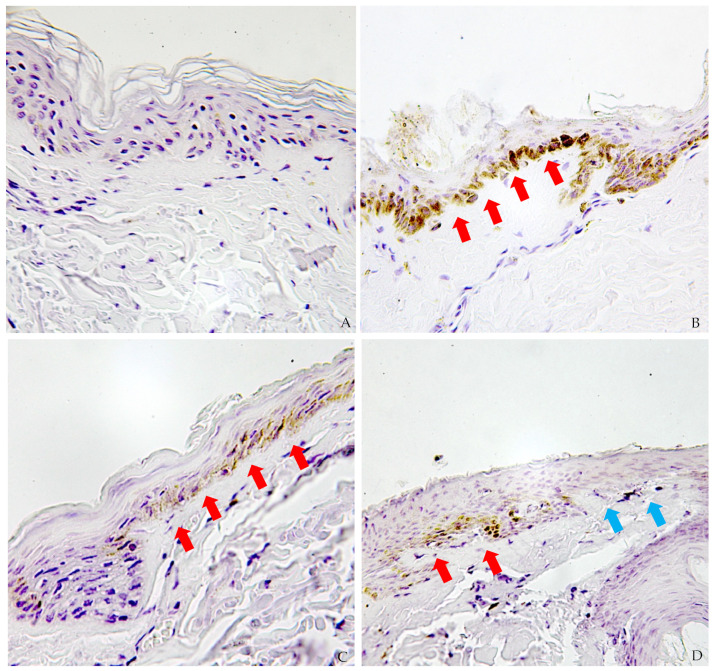
(**A**) Healthy skin, characterized by normal histomorphology; well-represented keratin layer. The anti-caspase-3 antibody reaction is negative. (40×) (**B**) Injured skin sample from the full loop of the hanging sulcus. Detachment of superficial layer is detected. Spinous and granular layers are poorly assessable. Positive reaction of the basal layer to immunohistochemistry with anti-caspase-3 antibody (40×) (red arrows). (**C**,**D**) Border between healthy skin and injured skin. The basal layer of the injured skin is clearly visible compared to the healthy skin (see below). (40×) (red arrows—positive cells, blue arrows—negative cells).

**Figure 2 ijms-26-03317-f002:**
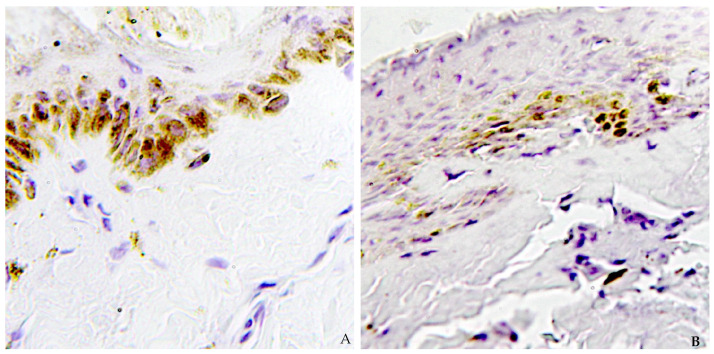
(**A,B**) A different pattern of expression was found. Granular cytoplasmatic pattern of caspase-3 expression was predominant. Nuclear and cytoplasmatic patterns were rare but present (40×).

**Figure 3 ijms-26-03317-f003:**
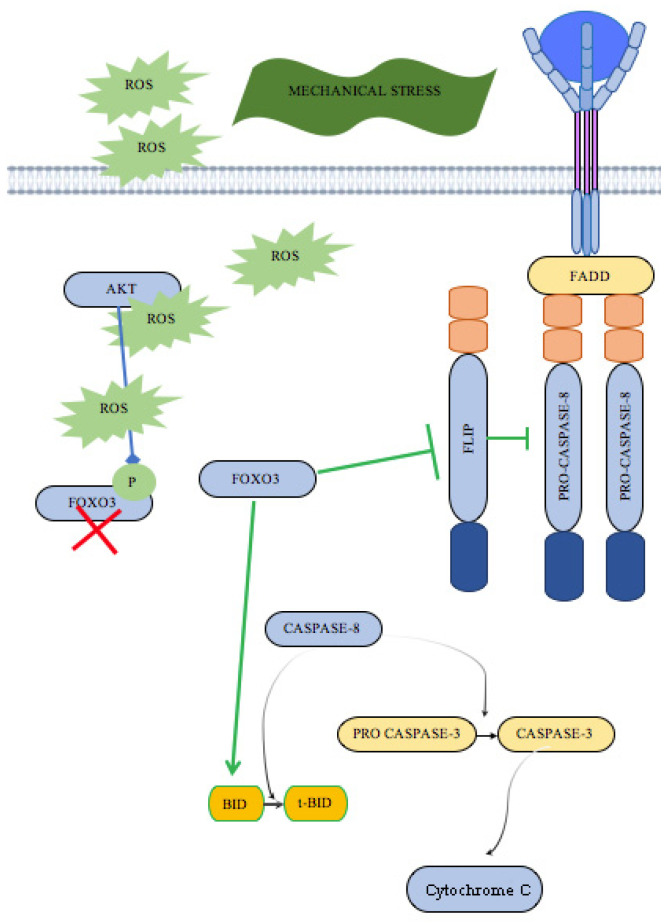
Protein interactome model of mechanical stress-induced apoptosis based on recent advances in hanging injury research.

**Table 1 ijms-26-03317-t001:** Semi-quantitative analysis of the immunohistochemical reaction of anti-caspase-3 on the skin of the hanging furrow.

n°	Sex	Ligature Mark Caspase-3	Uninjured Skin Caspase-3
1	M	3	0
2	F	2	1
3	M	2	0
4	M	3	0
5	M	3	0
6	M	2	1
7	F	2	0
8	F	2	0
9	F	3	0
10	M	3	0
11	M	2	1
12	F	3	0
13	M	2	0
14	M	3	0
15	M	3	0
16	F	2	1
17	M	2	0
18	M	2	0
19	M	2	1
20	F	3	0
21	M	3	0

**Table 2 ijms-26-03317-t002:** The amount and extent of marker expression were scored for each section from 0 to 3. The interpretation of our scoring system is shown in this table.

Grade	Interpretation
3	Positive staining
2	Minimal decrease in staining compared to normally stained tissue
1	A clear decrease in staining with some positivity (brown color) remaining
0	No positive staining

## Data Availability

The data presented in this study are available on request from the corresponding author.

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
