# Peer review of "Evaluation of Apoptotic Caspase-3 Immunopositivity in Human Model of Asphyxial Death"

_ijms, 2025, doi:10.3390/ijms26073317_

Round 1
Reviewer 1 Report
Comments and Suggestions for Authors
The paper reports a preliminary study carried out on autoptic skin samples from people who died by hanging compared to other causes of death, focusing on Caspase-3 immunoexpression as a marker of vitality.
The study design is well-developed and sound, and the results appear attractive, as they can impact the forensic sciences community (e.g., in the literature, many reports on vitality markers in ligature marks are present, but I did not find similar studies on Caspase-3).
I am also pleased with the explanation of the molecular mechanism of apoptosis under the research hypothesis, which provides a comprehensive context for the paper in the discussion section.
However, some revisions are mandatory. Please see below.
- Improving English grammar and expression before publication can be considered. Authors should get a professional language service to help with English expression and grammar. Alternatively, a native English speaker could revise the manuscript.
- Reference 6 in line 60, reference 6 in lines 63-68, reference 20 in line 118, reference 21 in lines 132, reference 22 in lines 136, references 24, 25, 26, 27 in lines 144-145, reference 29 in line 156, reference 16 in line 160, should be moved at the end of the tenses.
- In line 75, the authors mentioned, “… In cases of post-mortem cadaver suspension …”. I did not understand whether they tested autoptic tissues in those cases or if this was a citation from the literature. If they tested autoptic tissue in post-mortem suspension, it would add significant value by including Caspase-3 immunoexpression in those cases.
- Reference 22 in line 393 is missing the authors. Please address this.
Improving English grammar and expression before publication can be considered. Authors should get a professional language service to help with English expression and grammar. Alternatively, a native English speaker could revise the manuscript.
Author Response
Dear Reviewer 1,
The research team greatly appreciates your valuable comments and suggestions, which have helped us improve the quality of the manuscript revised by you.
[Reviewer 1]:
The paper reports a preliminary study carried out on autoptic skin samples from people who died by hanging compared to other causes of death, focusing on Caspase-3 immunoexpression as a marker of vitality.
The study design is well-developed and sound, and the results appear attractive, as they can impact the forensic sciences community (e.g., in the literature, many reports on vitality markers in ligature marks are present, but I did not find similar studies on Caspase-3).
I am also pleased with the explanation of the molecular mechanism of apoptosis under the research hypothesis, which provides a comprehensive context for the paper in the discussion section.
However, some revisions are mandatory. Please see below.
[Authors]:
We sincerely appreciate the reviewer’s recognition of our work. A comprehensive investigation into this specific mechanism of violent mechanical asphyxia can significantly aid in the accurate forensic diagnosis of complex cases, where the postmortem suspension of the body may obscure the actual cause of death and lead to its misinterpretation as a hanging.
[Reviewer 1]:
Improving English grammar and expression before publication can be considered. Authors should get a professional language service to help with English expression and grammar. Alternatively, a native English speaker could revise the manuscript.
[Authors]:
Thank you very much for your suggestion regarding the English revision by a native speaker. We have carefully reviewed the grammar and corrected any typographical errors while ensuring that the language remains clear and highly readable for the scientific community, prioritizing simple and concise phrasing.
We followed the aim of “authors should write not for themselves but for their readers.” [Bredan, Amin S, and Frans van Roy. “Writing readable prose: when planning a scientific manuscript, following a few simple rules has a large impact.” EMBO reports vol. 7,9 (2006): 846-9. doi:10.1038/sj.embor.7400800]
To achieve this, we shortened all unusually lengthy phrases. We also aimed to enhance the clarity of key concepts, making them more easily readable, especially in the introduction. You can find everything in the attached text.
[Reviewer 1]:
Reference 6 in line 60, reference 6 in lines 63-68, reference 20 in line 118, reference 21 in lines 132, reference 22 in lines 136, references 24, 25, 26, 27 in lines 144-145, reference 29 in line 156, reference 16 in line 160, should be moved at the end of the tenses.
[Authors]:
All the requested adjustments have been made, and the references have been moved to the end of the respective sentences as suggested.
[Reviewer 1]:
In line 75, the authors mentioned, “… In cases of post-mortem cadaver suspension …”. I did not understand whether they tested autoptic tissues in those cases or if this was a citation from the literature. If they tested autoptic tissue in post-mortem suspension, it would add significant value by including Caspase-3 immunoexpression in those cases.
[Authors]:
As mentioned in line 227, in the control group there are 8 cases of drug related deaths with 3 out 8 of these cases were suspended after death.
[Reviewer 1]:
Reference 22 in line 393 is missing the authors. Please address this.
[Authors]:
For some reason, the citation manager deleted authors from this reference, but we added them.
We sincerely appreciate your valuable feedback, which has helped enhance the quality of our manuscript.
Thank you for your time and consideration.
Best regards
Reviewer 2 Report
Comments and Suggestions for Authors
I believe this paper has the potential to become a top research article, especially if ligature materials are considered. I think there is relevant information on this topic available in the records, but that would take additional efforts. Including this data would enhance the discussion, strengthen the conclusion, and increase the overall attractiveness of the paper.
It is recommended to refer to hangings with the knot positioned at the back-right as “typical,” even though many sources categorize any back-positioned knot as “typical.” Following this guideline, this study ultimately aligns with existing literature, which indicates that only about one-third of hangings are classified as typical (specifically, 14.3% as reported by Chacko et al. in 2025).
Honestly, it is already sufficient for publication in its form of “communication,” with only minor revisions needed. However, the paper could be improved further, potentially warranting its rebranding as a full-length article. Regardless, these concerns should be addressed no matter what.
- The primary inclusion criterion is missing. You should clearly indicate which cohort the 21 cases were selected from—was it from the cohort of xx violent deaths? Please define the primary group for “randomization.” For example, "The autopsy databases from the Legal Medicine and Forensic Institutes of 'Sapienza' University of Rome, which included 500 reports, and the University of Foggia, with 300 reports..." Additionally, please specify the period during which the study was conducted.
- There is no need to separate each sentence into its own paragraph. For example, lines 207 to 216 are currently three distinct paragraphs. Lines 210 and 211 stand out, as they seem to belong to the following paragraph and should not be separated from it.
Author Response
Dear Reviewer2,
The research team sincerely appreciates your valuable comments and suggestions, which have significantly contributed to improving the quality of the revised manuscript.
[Reviewer 2]:
I believe this paper has the potential to become a top research article, especially if ligature materials are considered. I think there is relevant information on this topic available in the records, but that would take additional efforts. Including this data would enhance the discussion, strengthen the conclusion, and increase the overall attractiveness of the paper.
[Authors]:
We sincerely appreciate your positive assessment and insightful suggestion. We acknowledge the importance of considering ligature materials, as this could further enrich the discussion and strengthen our conclusions. While incorporating this data would require additional efforts, we will certainly explore the available records to assess its feasibility.
Thank you again for your valuable input, which has helped improve the quality and impact of our manuscript.
[Reviewer 2]:
It is recommended to refer to hangings with the knot positioned at the back-right as “typical,” even though many sources categorize any back-positioned knot as “typical.” Following this guideline, this study ultimately aligns with existing literature, which indicates that only about one-third of hangings are classified as typical (specifically, 14.3% as reported by Chacko et al. in 2025).
Honestly, it is already sufficient for publication in its form of “communication,” with only minor revisions needed. However, the paper could be improved further, potentially warranting its rebranding as a full-length article. Regardless, these concerns should be addressed no matter what.
[Authors]:
As you suggested, we have revised line 228 to include the following: “In 11 of the 21 cases, the individuals used wide, soft, and flexible materials like sheets for hanging, while in the other 10 cases, they used hard materials like ropes. 17 cases out of 21 was classified as typical hanging as reported from Chacko et al 2025[41]. In fact, the knot was positioned at the back of the neck in 7 cases, on the right side in 10 cases, and on the left side in 4 cases. The hanging was complete in 15 cases and incomplete in 6 cases.”.
[Reviewer 2]:
The primary inclusion criterion is missing. You should clearly indicate which cohort the 21 cases were selected from—was it from the cohort of xx violent deaths? Please define the primary group for “randomization.” For example, "The autopsy databases from the Legal Medicine and Forensic Institutes of 'Sapienza' University of Rome, which included 500 reports, and the University of Foggia, with 300 reports..." Additionally, please specify the period during which the study was conducted.
[Authors]:
We aimed to provide more precise details on the sample selection and study context to enhance the transparency of our methodology. Therefore, we have included a clear explanation of the selection criteria, randomization process, and study period.
“Between 2020 and 2024, the autopsy databases of the Legal Medicine and Forensic Institutes of Sapienza University of Rome and the University of Foggia recorded cases of autopsies for all causes of death, both violent and natural. From this dataset, cases were randomly selected from autopsies ordered by judicial authorities in cases of suspected criminal activity. Once anonymized, the database was categorized into violent and non-violent deaths. The violent death cases were analyzed, and a subset of individuals who died by hanging was extracted. The study group was then selected from this database through a random selection of 21 cases. The primary endpoint of the study was to assess the vitality of the ligature mark in all hanging cases, regardless of the knot position. The same procedure was applied to select the control cases.”
[Reviewer 2]:
There is no need to separate each sentence into its own paragraph. For example, lines 207 to 216 are currently three distinct paragraphs. Lines 210 and 211 stand out, as they seem to belong to the following paragraph and should not be separated from it.
[Authors]:
We followed your advice and merged the requested paragraphs. Indeed, this has made the reading flow more smoothly.
We truly appreciate your insightful feedback, which has contributed to improving the quality of our manuscript.
Thank you for your time and thoughtful review.
Best regards
Round 2
Reviewer 2 Report
Comments and Suggestions for Authors
I believe this paper makes a valuable contribution to the literature in its current form. Even though the authors agree with my observation that the paper holds even greater potential, it is reasonable to accept the current version and encourage the submission of future papers.